# Predictive Factors for Onset of Moderate-to-Severe Disability Following Hospital Discharge Due to Rib Fractures

**DOI:** 10.3390/healthcare12191984

**Published:** 2024-10-04

**Authors:** María Granados Santiago, Laura López López, Florencio Quero Valenzuela, Andrés Calvache Mateo, Javier Martín Núñez, Araceli Ortiz Rubio, Marie Carmen Valenza

**Affiliations:** 1Department of Nursing, Faculty of Health Sciences, University of Granada, 18071 Granada, Spain; mariagranados@ugr.es; 2Department of Physiotherapy, Faculty of Health Sciences, University of Granada, 18071 Granada, Spain; lauralopez@ugr.es (L.L.L.); andrescalvache@ugr.es (A.C.M.); javimn@ugr.es (J.M.N.); cvalenza@ugr.es (M.C.V.); 3Thoracic Surgery Department, Hospital Virgen de las Nieves de Granda, 18071 Granada, Spain; fquero@ugr.es

**Keywords:** rib fractures, length of stay, hospitalization, disability, quality of life

## Abstract

Background: According to previous studies, a prolonged hospital stay, along with the patient’s clinical features, can lead to the onset of disability. Objectives: This study aimed to identify predictive factors of moderate-to-severe disability following hospital discharge in rib fracture patients. Methods: We conducted a retrospective study with hospitalized adult patients with rib fractures who received conservative treatment. Patients’ clinical profiles and characteristics were collected from their clinical histories and healthcare professional records. Results: Overall, patients exhibited a 44% rate of moderate-to-severe disability after a six-day hospital stay. The incidence of patients with a prolonged hospital stay who showed disabilities was associated with male sex (HR 0.73, *p* < 0.001), presence of first rib fracture (HR 1.78, *p* = 0.047), presence of flail chest (HR 1.29, *p* = 0.046), severity of lung injury (HR 1.65, *p* < 0.001), and functional status (HR 1.37, *p* < 0.001). Conclusions: Moderate-to-severe disability in patients with rib fractures may depend on factors such as sex, functionality, severity of lung injury, and presence of first rib fracture and flail chest during a prolonged hospital stay.

## 1. Introduction

Rib fractures are a severe consequence of blunt chest trauma, with a prevalence upwards of 10–20% [1,2]. They are most frequently caused by adult falls (64%) and motor vehicle crashes (27%) [3].

The therapeutic management of rib fractures usually involves two options: analgesia and respiratory support, if required [4,5], and surgical stabilization with internal fixation [6,7]. Regardless of the chosen management strategy, patients with three or more fractured ribs should require hospitalization [8].

The length of hospital stay for patients with rib fractures can vary from the short to long term, depending on the clinical evolution of the patient’s condition [9]. The early treatment priority is to achieve a short hospital stay with low morbidity [10], which leads to significant health cost savings [11]. However, increased age and injury severity are associated with higher rates of hospital complications and mortality, thereby extending the length of hospital stay [12,13,14]. In addition to the immediate impact of rib fractures, patients may report long-term complications after hospital discharge, such as fracture non-union, intercostal nerve entrapment, chronic pain, deformity, and disability [15,16,17].

A prolonged hospital stay in the context of severe rib fractures can advance the onset of disability [18]. After hospital discharge, up to 53% of rib fracture patients showed long-term disability, significantly affecting their quality of life [19]. Despite these findings, there is a lack of studies evaluating the relationship between the background characteristics of rib fracture patients and the development of moderate-to-severe disability following hospital discharge.

In this study, we hypothesized that the duration of hospital stay and clinical features of rib fracture patients are associated with moderate-to-severe disability following hospital discharge. Identifying these predictive factors could facilitate early intervention to prevent the onset of disability in this population. Thus, we aimed to identify the predictive factors of moderate-to-severe disability following hospital discharge in patients with rib fracture.

## 2. Materials and Methods

### 2.1. Study Design and Setting

A retrospective observational study of rib fracture patients was performed. All participants were hospitalized in the hospital’s Thoracic Surgery Service. The study was approved by the Clinical Research Ethics Committee of the hospital (1088-N-15). The RECORD statement for studies conducted using routinely collected health data was followed [20].

### 2.2. Participants

All enrolled patients gave informed consent to participate in the study and to the processing of their clinical data. This study included adult patients with rib fractures who received conservative treatment during their hospital stay. These patients may have had rib fractures either isolated or combined with other traumatic injuries. The exclusion criteria were patients who received surgical treatment, had prior traumatic injuries or rib fractures, and/or were unable to provide informed consent. Patients receiving mechanical ventilatory support or requiring ICU admission were also excluded.

All participants received the hospital protocol for conservative treatment of rib fracture to stabilize the pain and lung function during the hospital stay. Systemic analgesia, anti-inflammatory drugs, and myorelaxants were administered for pain relief. Furthermore, patients could receive regional anesthetic techniques (intercostal and paravertebral blocks), and/or epidural analgesia according to each patient’s condition. Regarding lung function, expectorant aerosolized medications were provided to help patients with cough expectoration drainage. The aim was to improve the oxygenation and ventilation abnormalities and to prevent lung infection. If the patients had low oxygen levels, oxygen was provided with a nasal catheter or oxygen mask.

### 2.3. Data Collection

Data for this study were obtained from the patients’ clinical histories and healthcare professional records from hospital admission to discharge. Specifically, the length of hospital stay and clinical features of rib fracture patients were collected. An investigator conducted a manual chart review to gather the data.

Demographic and clinical data were gathered for all rib patients including age, sex, comorbidities, rib fracture causes, and pharmacotherapy. The comorbidities were evaluated according to the Charlson Comorbidity Index [21]. A higher score represents more severe illness impacts.

For the determination of rib fracture pattern, the number of ribs fractured and the presence of bilateral rib fractures, flail chest, additional fractures (clavicle, sternum, scapular, or transverse process), and first rib fracture were collected.

The rib fracture severity in blunt trauma was evaluated using a chest trauma score [22], and a lung injury score [23] was used to measure the lung injury severity and development of respiratory failure derived from a rib fracture. The chest trauma score includes four domains with a point system assigned: age (<45 years = 1, 45–65 = 2, >65 = 3), pulmonary contusion (none = 0, unilateral minor = 1, bilateral minor = 2, unilateral major = 3, bilateral major = 4), number of rib fractures (<3 = 1, 3–5 = 2, >5 = 3), and the presence of bilateral rib fractures. This score ranges from 2 (low severity) to 12 (greater severity). The lung injury score is composed of four components: the extent of pulmonary densities on chest radiography (no alveolar consolidation = 0; alveolar consolidation confined to one quadrant = 1; alveolar consolidation confined to two quadrants = 2; alveolar consolidation confined to three quadrants = 3; alveolar consolidation confined to four quadrants = 4), gas exchange abnormalities (PaO2/FiO2 ratio), the positive end-expiratory pressure (PEEP) score (if patients are ventilated), and lung compliance (when available). The total value is obtained by dividing the aggregate sum by the number of components used, with no lung injury having a score of 0, mild to moderate injury in a range of 0.1 to 2.5, and severe injury > 2.5.

The Katz Index of Activities of Daily Living (ADLs) [24] was used to measure the disability of the people in six ADLs: bathing, dressing, going to the toilet, transference, continence, and feeding. The observer scores every activity on a scale of 2 levels (1 = dependence; 0 = independence). The absence of disability has a 0 score, with mild disability 1, moderate disability in a range of 2–3, and severe disability 4–6.

All outcome measures were recorded at hospital admission, with the Katz Index of Activities of Daily Living assessed both at pre-injury and at the time of hospital discharge.

### 2.4. Statistical Analysis

All rib fracture patients were allocated into two groups according to the length of hospital stay. The cut-off point was established based on the article by Dalton et al. (2019) [11], where three days of hospitalization were established for the recovery of patients with a rib fracture. In summary, a short hospital stay was a hospitalization time ≤ 3 days, and a long hospital stay was a hospitalization time > 3 days.

The characteristics of rib fracture patients were presented as the mean ± SD or percentage, as appropriate. Data were checked for normality before statistical analysis. The categorical variables were compared by contingency tables. The continuous variables were compared using an independent samples *t*-test (*t*-statistics, *p*-value). The statistical analysis was established at a 95% confidence level. A *p*-value of less than 0.05 was considered statistically significant.

A Kaplan–Meier analysis was used to graphically represent the probability of having moderate-to-severe disability at discharge for all rib fracture patients. To identify risk factors of moderate-to-severe disability following hospital discharge, we estimated cause-specific hazard ratios (HRs) with 95% confidence intervals in a separate analysis, as calculated using univariable Cox regression models. The Kaplan–Meier method with the log-rank test was used for statistical comparison between lung injury and polypharmacy (more than three drugs per day).

In these analyses, all outcome measures of patients with rib fractures in hospital admission were included. The threshold for statistical significance was set at a *p*-value of 5% (*p* < 0.05). All statistical analyses were conducted using IBM SPSS Statistics (Version 23.0, IBM Corp., Armonk, NY, USA) by two investigators.

*p* values were adjusted for multiple comparisons, when indicated, with Bonferroni correction. A Bonferroni corrected *p* value < 0.05 was considered statistically significant.

## 3. Results

Finally, 197 rib fracture patients without surgery were included in the study (Figure 1).

The patient’s baseline characteristics are shown in Table 1.

Both groups were almost similar at baseline. The median age was 62 years old in patients with a short hospital stay, and 59 years in patients with a long hospital stay. Within the short hospital stay group, there was 26.7% of male, being 27.5% of patients with a long hospitalization time. The pharmacotherapy showed statistical differences, highlighting a greater number of drugs in patients with a long hospital stay versus patients with a short hospital stay (mean of 4.4 vs. 2.5, *p* = 0.035). The rib fracture pattern did not show statistical differences between groups (*p* > 0.05). Additionally, statistical differences were observed between groups in the rib fracture severity, highlighting a higher severity related to lung injury in patients with a long hospitalization time (*p* < 0.007). The pre-injury disability did not find statistical differences between groups (*p* > 0.05).

Overall, hospitalized patients with rib fractures revealed 3, 6, 10, and 15 days overall moderate-to-severe disability rates of 7%, 44%, 81%, and 94%, respectively (Figure 2). Median survival for the rib fracture patients who had moderate-to-severe disability after hospital discharge was 8 days.

Additionally, patients with lung injury and a consumption of three or more drugs have a longer hospital stay (*p* = 0.003) (Figure 3).

The univariable Cox regression analysis was shown in Table 2.

Any of the measured outcomes were able to predict disability severity for the short hospital time group (*p* > 0.05).

The incidence of patients with a long hospitalization time who showed moderate–severe disability was associated with male sex (hazard ratio [HR] 0.73, 95% CI 0.61–0.87, *p* < 0.001), presence of first rib fracture and flail chest (HR 1.78, 95% CI 0.71–4.44, *p* = 0.047; HR 1.29, 95% CI 1.01–1.65, *p* = 0.046, respectively), severity of lung injury (HR 1.65, 95% CI 0.44–0.93, *p* < 0.001), and pre-injury disability (HR 1.37, 95% CI 1.16–1.62, *p* < 0.001).

## 4. Discussion

This study identified several predictive factors for moderate-to-severe disability following hospital discharge in rib fracture patients. Factors such as sex, pre-injury disability, severity of lung injury, and the presence of a first rib fracture or flail chest were associated with moderate-to-severe disability in patients with prolonged hospitalization.

Previous studies have reported similar profiles (age, sex, etc.) in rib fracture patients involving conservative treatment [25,26]. Our study showed a greater medication intake in rib fracture patients with a longer hospitalization. Various previous studies observed that polypharmacy was not related to the length of hospital stay. However, the adverse effects were a significant predictor of an increase in hospital stay by nearly 4 days [27].

Rib structural characteristics decline over the years and may change according to sex. Bone demineralization and abnormal hormone levels in females begin in middle age as consequences of menopause [28]. These factors could contribute to a greater rate of rib fractures in females compared to males and frequently demand longer hospital lengths of stay [29,30,31]. Thus, our study showed sex as a predictive factor of disability after discharge, with male sex being a factor that reduces the risk of disability after longer hospitalization.

A prolonged hospital stay can lead to physical and functional impairments after hospital discharge due to factors such as cognitive state, level of independence, drug effects, nutritional status, post-traumatic pain, type of injury, trauma score, and injury severity [32,33]. Additionally, demographic variables, including age, sex, and race/ethnicity, have been identified as non-modifiable factors associated with prolonged hospitalization [34]. Pre-existing comorbidities in trauma patients have also been linked to longer hospital stays [35]. Multiple studies observed the presence of long-term complications after a flail chest injury, including pain, respiratory problems, and prolonged disability in different degrees [16,19,36,37]. These findings are in the same line as our study, where over 80% of patients suffer a flail chest after rib fractures, being a predictor of moderate-to-severe disability in longer hospitalization. Additionally, fractures of the first rib are associated with the severity of clinical status of patients, a higher incidence of further injuries, and an increased morbidity [38,39].

More than a third of patients have rib fractures with underlying lung injury [40]. Multiple complications related to lung injury are associated with rib fractures, including pneumonia, hemothorax, pneumothorax, and acute respiratory distress syndrome (ARDS) [41]. As our research showed, the presence of lung injury may increase the hospital stay and long-term disability [42]. Hence, prior studies suggested a multidisciplinary team of nurse practitioners, physiotherapists, and occupational therapists decreased the use of the ventilatory approach and hospital lengths of stay [17,43].

Multiple studies used the time of recovery to predict future complications in the presence of a chronic disease [44], or after surgery [45]. According to these studies, our research observed the timing of recovery in the acute care of patients with rib fractures to prevent the presence of disability after hospital discharge.

Several limitations were identified in our study. First, a retrospective design limited the data collection, and patients could not be followed up after hospital discharge. Secondly, this study was limited to one hospital, which could result in potential selection bias. Thirdly, the study design makes it difficult to confirm the predictive factors of disability because there is no control for confounding factors. Additionally, one investigator conducted a manual chart review of the patient records, so we were unable to calculate the Kappa analysis. Furthermore, the measurement of the lung injury score was affected by the unavailability of certain components in the patient’s clinical history and the lack of detailed procedure records. There was a limitation with the data collected from the hospital records: we could not record it in a systematic way, and only one investigator was able to collect the data. Additionally, we were unable to follow up on the disability status after hospital discharge. Future studies should include a multicentric design with patients of different ethnic backgrounds. Finally, future longitudinal studies are recommended to confirm our results and to assess the long-term outcomes after rib fractures, as well as to conduct multivariable analyses for a more comprehensive understanding of the effects of different variables.

## 5. Conclusions

The main finding of our study is that a prolonged hospital stay predicts moderate-to-severe disability in rib fracture patients, depending on factors such as sex, pre-injury disability, the severity of lung injury, and the presence of first rib fracture and flail chest.

## Figures and Tables

**Figure 1 healthcare-12-01984-f001:**
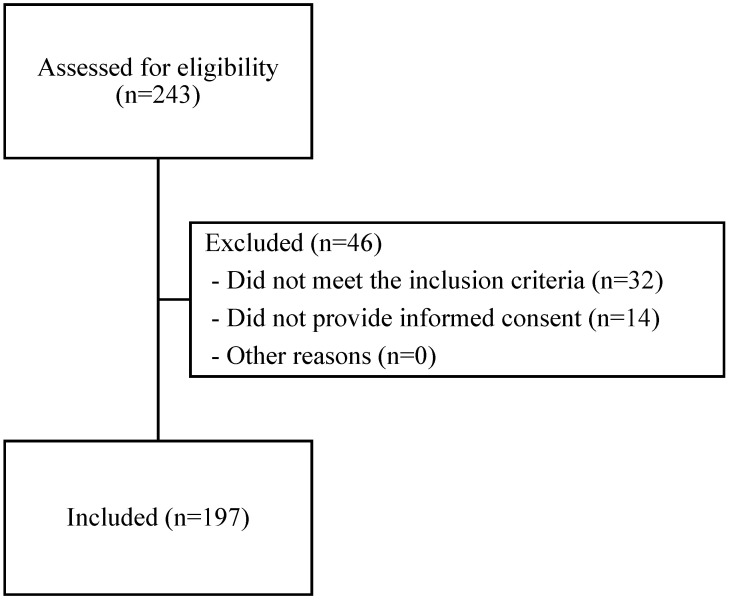
Flowchart.

**Figure 2 healthcare-12-01984-f002:**
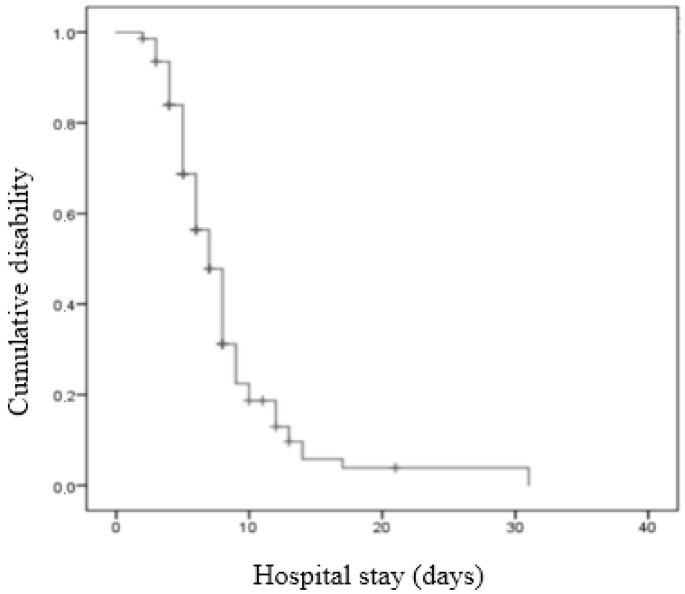
Kaplan–Meier curve for rib fracture patients after hospital discharge.

**Figure 3 healthcare-12-01984-f003:**
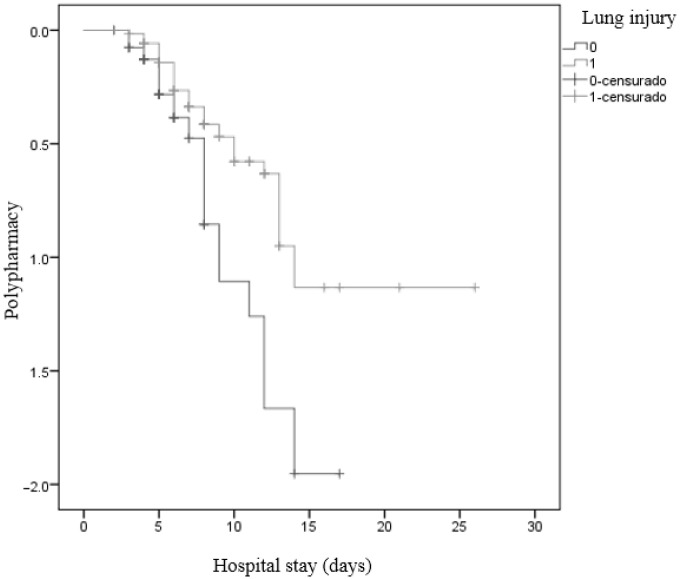
Kaplan–Meier curve and long rank for lung injury and polypharmacy.

**Table 1 healthcare-12-01984-t001:** Characteristics of patients with rib fractures in hospital admission.

	Outcomes Measures	Short Hospital Stay (*n* = 15)	Long Hospital Stay (*n* = 182)	*p*-Value
General characteristics	Age (years)	62.27 ± 16.25	59.77 ± 17.80	0.600
	Sex, *n* (male%)	4 (26.7)	50 (27.5)	0.946
	Comorbidities	1.71 ± 1.82	1.65 ± 1.89	0.908
Rib fracture causes	Fall, *n* (%)	8 (53.3)	100 (55.2)	0.886
	Traffic, *n* (%)	7 (46.7)	79 (43.4)	0.807
Pharmacotherapy	No. drugs	2.5 ± 3.06	4.40 ± 5.69	0.035 *
Rib fracture pattern	No. rib fractures	4.53 ± 2.06	5.20 ± 2.55	0.324
	First Rib Fractured, *n* (%)	2 (13.3)	20 (11.3)	0.814
	Bilateral rib fractures, *n* (%)	3 (18.2)	29 (16)	0.848
	Flail chest, *n* (%)	13 (86.7)	149 (82)	0.652
	Clavicle fracture, *n* (%)	0	22 (12.2)	0.152
	Sternum fracture, *n* (%)	2 (13.3)	12 (6.6)	0.333
	Scapular fracture, *n* (%)	1 (6.7)	9 (5)	0.774
	Transverse process fracture, *n* (%)	0	7 (3.9)	0.438
Rib fracture severity	Chest trauma score	4.60 ± 1.21	4.94 ± 1.58	0.421
	Lung injury score	0.22 ± 0.41	0.64 ± 0.57	0.007 *
Pre-injury disability	Katz	1.93 ± 1.58	2.09 ± 1.59	0.711

* Results are expressed as mean ± SD. * *p* < 0.05.

**Table 2 healthcare-12-01984-t002:** Hazard ratios for disability by Cox regression model in patient background.

		Short Hospital Stay		Long Hospital Stay	
	Outcomes Measures	Hazard Ratio (95% CI)	*p* Value	Hazard Ratio (95% CI)	*p* Value
General characteristics	Age (years)	0.98 (0.94–1.02)	0.361	0.99 (0.98–1.01)	0.526
	Sex (male)	0.62 (013–2.99)	0.552	0.73 (0.61–0.87)	<0.001 **
	Comorbidities	0.77 (0.49–1.18)	0.229	1.04 (0.95–1.14)	0.251
Rib fracture causes	Fall	1.49 (0.39–5.54)	0.554	0.96 (0.83–1.11)	0.569
	Traffic	0.67 (0.18–2.51)	0.554	1.03 (0.89–1.19)	0.673
Pharmacotherapy	No. drugs	0.96 (0.85–1.08)	0.553	1.01 (0.93–1.08)	0.812
Rib fracture pattern	No. rib fractures	0.94 (0.68–1.29)	0.699	0.98 (0.89–1.08)	0.164
	First Rib Fractured	0.91 (0.01–1.45)	0.09	1.78 (0.71–4.44)	0.047 *
	Bilateral rib fractures	0.19 (0.06–0.55)	0.464	0.87 (0.69–1.09)	0.227
	Flail chest	0.04 (0.01–1.62)	0.781	1.29 (1.01–1.65)	0.046 *
	Clavicle fracture	-	-	1.17 (0.94–1.46)	0.14
	Sternum fracture	0.49 (0.1–2.37)	0.376	1.01 (0.72–1.39)	0.96
	Scapular fracture	-	-	0.88 (0.58–1.33)	0.564
	Transverse process fracture	-	-	1.54 (0.93–2.55)	0.47 *
Rib fracture severity	Chest trauma score	0.87 (0.51–1.49)	0.629	0.96 (0.84–1.09)	0.444
	Lung injury score	0.9 (0.1–11.12)	0.336	1.65 (0.44–0.93)	<0.001 **
Pre-injury disability	Katz	1.47 (0.79–2.74)	0.216	1.37 (1.16–1.62)	<0.001 **

* *p* < 0.05, ** *p* < 0.001.

## Data Availability

No additional data are available.

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
