# Peer review of "Predictive Factors for Onset of Moderate-to-Severe Disability Following Hospital Discharge Due to Rib Fractures"

_healthcare, 2024, doi:10.3390/healthcare12191984_

Round 1
Reviewer 1 Report
Comments and Suggestions for Authors I found it very interesting, because it addresses a very common problem that we as thoracic surgeons often encounter: chronic pain after rib fractures. Taking into account the fact that, at least in our country, rib fixation is applied to a minimal percentage of cases, the problematic of pain is more common that we thought it was, and i think this paper has good importance to practitioners. That is why I agree with the publishing of the paper.
At table 1 there are three columns Entitled Title 3 and Title 4 (twice) maybe they had other column names.
Author Response
Comment 1: I found it very interesting, because it addresses a very common problem that we as thoracic surgeons often encounter: chronic pain after rib fractures. Taking into account the fact that, at least in our country, rib fixation is applied to a minimal percentage of cases, the problematic of pain is more common that we thought it was, and i think this paper has good importance to practitioners. That is why I agree with the publishing of the paper.
At table 1 there are three columns Entitled Title 3 and Title 4 (twice) maybe they had other column names.
Response 1: Thank you for considering our study for publication and highlighting its relevance to clinical practice. We agree with this comment and have included additional titles for the table columns.
Reviewer 2 Report
Comments and Suggestions for Authors
Some questions arise:
Basic:
- Does a longer hospital stay really contribute to higher disability? Is the disability not determined by the severity of initial trauma that leads to longer hospital stay?
- In the conclusion -if i understand it correctly - it seems like the prolonged hospital stay is simply a retrospective finding. It would be rather of meaning to identify such patients in the first place and why they are linked to higher disability? What is the overall message? More severe injuries are linked to longer hospital stay or did you evaluate equally severe injuries and try to define which patients had more troubles than others? What ever the message is this should be stated more clearly throughout the entire manuscript!
- What is the explanation that a longer hospital stay would lead to higher disability?
Specific:
- Do patients with longer hospital stay have more severe trauma/injuries and combination of listed regions/injuries. Add numbers to the percentages when having more than 100 patients which is applicable. One might think that a longer hospital stay is linked to more combined injuries? Is that the case in this study?
- Are those isolated first rib fractures? Usually those are signs of higher force therefore combined with other injuries?
- Is male sex connected to more severe trauma/injury?
- Fig. 2 what is meant with median survival? I assume all of them were again discharged?
- How was the diagnostics performed plain radiographs or CT? Especially pulmonary contusion? And how was its quantification in minor and major? Who and how many evaluated the quantification?
- If patients were treated conservatively and assumingly not ventilated how accurate is the lung injury score as described? How and by whom was the extent of pulmonary density measured? How the lung compliance and in how many patients?
- How was the data extraction performed? Are there missing data and what did I you do about it? How many investigators were included? Who collected the retrospective data, and how did they collect it? Was it manual chart review? How many people collected the data? If this was a manual chart review, please include a Kappa analysis across different data extractors.
- Limitations: only 2 limitations? In that case how was the data collection limited? What else would authors have liked to present or gain information about that could further specify that issue? what potential selection bias are authors referring to?
Statistics:
- First table: 17 p-values and no statistical correction which is mandatory!
- Table 1: title 3 and 4 are missing! Further how many patients per group (also not detectable in the text)? How many patients with combined injuries/isolated “minor” injuries.
- Second table again over 30 p-values… Correction?
The missing statistical correction makes the represented methodology questionable!
Kind regards
Comments on the Quality of English Languageminor expressions throughout the manuscript such as
rib fractures patients
or
line 137 The median age was 62 years old in patients with a short hospital stay, and 59 years old in patients ....omit "old"
Author Response
Some questions arise:
Basic:
Comment 1: Does a longer hospital stay really contribute to higher disability? Is the disability not determined by the severity of initial trauma that leads to longer hospital stay?
Response 1: Thank you for your comment. In the data collected at hospital admission for the study, we did not find any differences in rib fracture patterns or severity between short and long hospital stays, except for the lung injury score. There was an error in Table 1, which we have corrected to clarify this point.
Comment 2: In the conclusion -if i understand it correctly - it seems like the prolonged hospital stay is simply a retrospective finding. It would be rather of meaning to identify such patients in the first place and why they are linked to higher disability? What is the overall message? More severe injuries are linked to longer hospital stay or did you evaluate equally severe injuries and try to define which patients had more troubles than others? What ever the message is this should be stated more clearly throughout the entire manuscript!
Response 2: Agreed. We evaluated patients who received conservative treatment but had different durations of hospital stay to identify factors influencing disability at hospital discharge. The main purpose of our study was to identify factors that could impact the onset of disability at discharge, comparing the characteristics of patients with short and long hospital stays, including pre-injury disability. We found no significant differences at hospital admission and observed the factors contributing to the onset of disability, independent of rib fracture severity or pattern. We have revised the manuscript accordingly to emphasize this point.
Comment 3: What is the explanation that a longer hospital stay would lead to higher disability?
Response 3: Thank you for pointing this out. A prolonged hospital stay can lead to physical and functional impairments after hospitalization due to factors such as cognitive state, level of independence, drug effects, nutritional status, post-traumatic pain, type of injury, trauma score, and injury severity (Chona et al., 2017; Shafi et al., 2010). Additionally, demographic variables, including age, sex, and race/ethnicity, are considered non-modifiable factors associated with prolonged hospitalization (Lang et al., 2006). Pre-existing comorbidities in trauma patients have also been linked to prolonged stays (Brotemarkle et al., 2015). We have included in the discussion section.
Specific:
Comment 4: Do patients with longer hospital stay have more severe trauma/injuries and combination of listed regions/injuries. Add numbers to the percentages when having more than 100 patients which is applicable. One might think that a longer hospital stay is linked to more combined injuries? Is that the case in this study?
Response 4: According to Table 1, our study did not find differences in rib fracture patterns or severity, except for the lung injury score. We agree with this comment and have included the numbers alongside the percentages.
Comment 5: Are those isolated first rib fractures? Usually those are signs of higher force therefore combined with other injuries?
Response 5: Thank you for pointing this out. Patients may have had rib fractures either isolated or combined with other traumatic injuries. We have modified the methods section to reflect this.
Comment 6: Is male sex connected to more severe trauma/injury?
Response 6: It is possible that the prevalence of certain comorbidities influences injury and varies by sex. However, according to our study, male sex is a protective factor against the onset of disability after prolonged hospitalization. We did not evaluate the relationship between sex and trauma severity.
Comment 7: Fig. 2 what is meant with median survival? I assume all of them were again discharged?
Response 7: Thank you for your comment. We have modified the figure, which now shows the relationship between the cumulative disability of patients with rib fractures and the length of hospital stay.
Comment 8: How was the diagnostics performed plain radiographs or CT? Especially pulmonary contusion? And how was its quantification in minor and major? Who and how many evaluated the quantification?
Response 8: Agree. The appearance of opaque or dense areas, as well as abnormal opacity or density in the lungs, which may represent consolidations, atelectasis, pulmonary edema, pleural effusions, among others, were considered. However, there was a lack of detailed procedural records to obtain the lung injury data. We followed the interpretation of the detailed components of the lung injury score. The information was collected from patient medical records with input from multiple physicians. We have provided more details about the score for clarification in the methods section and have included the lack of detailed lung procedure records as a limitation of our study.
Comment 9: If patients were treated conservatively and assumingly not ventilated how accurate is the lung injury score as described? How and by whom was the extent of pulmonary density measured? How the lung compliance and in how many patients?
Response 9: Thank you for pointing this out. This is a retrospective observational study, and we only included the components available in the patients' clinical records to calculate the lung injury score, with chest radiography and gas exchange abnormalities being more frequently recorded. We have accordingly included this as a limitation of our study.
Comment 10: How was the data extraction performed? Are there missing data and what did I you do about it? How many investigators were included? Who collected the retrospective data, and how did they collect it? Was it manual chart review? How many people collected the data? If this was a manual chart review, please include a Kappa analysis across different data extractors.
Response 10: Thank you for your comment. We included only patients with complete data. The physician from the hospital department conducted the manual chart review of the medical records. Afterward, two investigators analyzed the data. We were unable to conduct the Kappa analysis because only one person collected the data. We have included this as a limitation of the study and clarified it in the methods section.
Comment 11: Limitations: only 2 limitations? In that case how was the data collection limited? What else would authors have liked to present or gain information about that could further specify that issue? what potential selection bias are authors referring to?
Response 11: Agree. We have included additional limitations identified during the review to emphasize this point. Specifically, we noted that the data collected from the hospital records could not be recorded systematically, and only one investigator was able to collect the data. Additionally, we were unable to follow up on the disability status after hospital discharge.
Statistics:
Comment 12: First table: 17 p-values and no statistical correction which is mandatory!
Response 12: Thank you for pointing this out. We have applied the Bonferroni correction for multiple comparisons. The revised results have been included, and the methods section has been amended to reflect this correction. An updated version of the table is not necessary.
Comment 13: Table 1: title 3 and 4 are missing! Further how many patients per group (also not detectable in the text)? How many patients with combined injuries/isolated “minor” injuries.
Response 13: We agree with this comment and have included additional titles for the table columns and the sample size in each group.
Comment 14: Second table again over 30 p-values… Correction?
Response 14: As the before response 12. We have applied the Bonferroni correction for multiple comparisons. An updated version of the table is not necessary
Comment 15: The missing statistical correction makes the represented methodology questionable!
Response 15: Agree, we resolved the mistake, and we include our suggestions to improve the statistical analysis.
Kind regards
Comments on the Quality of English Language:
minor expressions throughout the manuscript such as “rib fractures patients” or line 137 “The median age was 62 years old in patients with a short hospital stay, and 59 years old in patients ....omit "old"
Response: Agree. We modified all text with your comments.
Reviewer 3 Report
Comments and Suggestions for Authors
In Table 1, the last 3 columns do not have appropriate labels. As mentioned in the discussion, this study is limited due to the observational nature. It is difficult to conclude that any of the variables are true predictive factors of disability as there is no control for any confounding factors.
Comments on the Quality of English LanguageOverall the quality was acceptable. In line 62, it should be “exclusion criteria”.
Author Response
Comment 1: In Table 1, the last 3 columns do not have appropriate labels. As mentioned in the discussion, this study is limited due to the observational nature. It is difficult to conclude that any of the variables are true predictive factors of disability as there is no control for any confounding factors.
Response 1: We agree with this comment and have added more descriptive titles for the table columns. Additionally, we have included the inability to confirm the predictive factors as a limitation of the study.
Comments on the Quality of English Language: Overall the quality was acceptable. In line 62, it should be “exclusion criteria”.
Response: Agree, we have revised and corrected the text.
Reviewer 4 Report
Comments and Suggestions for Authors
The publication analyzes the very important problem of the consequences of blunt chest trauma.
The authors draw attention to the fact that as many as 44% of moderate to severe disabilities occurred in patients hospitalized due to rib fractures.
The authors analyzed the clinical histories of patients hospitalized in the Department of Thoracic Surgery. In this work, they used appropriate inclusion and exclusion criteria for analysis. Only patients treated conservatively were included. Two main directions were analyzed - length of stay and clinical features of patients with rib fractures.
The study itself was well planned and conducted.
The Katz Index of Activity of Daily Living-ADL was used to measure disability in six groups. On a two-point scale. This scale was assessed upon admission to hospital and after discharge.
Rich material - 197 patients included in the analysis were analyzed statistically.
The cut-off point was hospitalization time above 3 days and below 3 days according to Dalton's article.
The probability of moderate or severe disability is graphically presented very nicely using the Kaplan-Meier analysis.
All statistical analysis was conducted reliably using IBM SPSS Statistics.
The results were presented in the correct tables.
A rich discussion and a large number of references are the strengths of this publication, which shows that gender is a predictor of disability after discharge from hospital, to the detriment of women.
It should be emphasized that the authors also presented the limitations of the work - the time of observation of patients after discharge and the fact that the work was limited to one hospital.
Rich references - 41 items and most of them published after 2015 are a big advantage of this work.
The work is valuable and contributes a lot to science.
Author Response
The publication analyzes the very important problem of the consequences of blunt chest trauma.
The authors draw attention to the fact that as many as 44% of moderate to severe disabilities occurred in patients hospitalized due to rib fractures.
The authors analyzed the clinical histories of patients hospitalized in the Department of Thoracic Surgery. In this work, they used appropriate inclusion and exclusion criteria for analysis. Only patients treated conservatively were included. Two main directions were analyzed - length of stay and clinical features of patients with rib fractures.
The study itself was well planned and conducted.
The Katz Index of Activity of Daily Living-ADL was used to measure disability in six groups. On a two-point scale. This scale was assessed upon admission to hospital and after discharge.
Rich material - 197 patients included in the analysis were analyzed statistically.
The cut-off point was hospitalization time above 3 days and below 3 days according to Dalton's article.
The probability of moderate or severe disability is graphically presented very nicely using the Kaplan-Meier analysis.
All statistical analysis was conducted reliably using IBM SPSS Statistics.
The results were presented in the correct tables.
A rich discussion and a large number of references are the strengths of this publication, which shows that gender is a predictor of disability after discharge from hospital, to the detriment of women.
It should be emphasized that the authors also presented the limitations of the work - the time of observation of patients after discharge and the fact that the work was limited to one hospital.
Rich references - 41 items and most of them published after 2015 are a big advantage of this work.
The work is valuable and contributes a lot to science.
Global response: Thank you very much for taking the time to review this manuscript. We appreciate all your detailed feedback and have improved the manuscript based on the reviewer's comments.